# Hypovitaminosis D and Aging: Is There a Role in Muscle and Brain Health?

**DOI:** 10.3390/nu12030628

**Published:** 2020-02-27

**Authors:** Patrizia D’Amelio, Luca Quacquarelli

**Affiliations:** 1Service de Gériatrie et de Réadaptation Gériatrique, CHUV, 1011 Lausanne, Switzerland; 2Department of Internal Medicine, Geriatric and Bone Disease Unit, University of Torino, 10126 Torino, Italy; luca.quacquarelli@unito.it

**Keywords:** vitamin D, brain, muscle, aging

## Abstract

The older-adult population is constantly increasing, hence aging and mechanisms leading to aging are a topic raising increasing interest. Hypovitaminosis D is common amongst old patients and has been proposed as causative of several chronic diseases. Here we review the role of hypovitaminosis D and vitamin D supplementation in sarcopenia and dementia, from bench to bedside.

## 1. Introduction 

Due to an increased life expectancy, aging and the mechanisms leading to it have become a pivotal issue in scientific research. Together with innocuous and physiological changes due to aging, there are pathological changes that increase the risk of disease, disability, or death linked to the development of frailty and chronic diseases, hence older adults can be divided into the fit and the frail.

During aging, all systems and organs are physiologically reduced in their function. However, identifying the risk factors leading to “fit” or “frail” aging is fundamental in order to understand the physiopathological mechanisms leading to frailty and suggest preventive measures. Health maintenance in older age is one the most important challenges for future medicine. Hence, ameliorating the burden of chronic disease in the elderly is also one of the goals of future medicine. Among the chronic conditions that severely affect patient quality of life, diseases affecting mobility and cognition—namely, sarcopenia and dementia—are the most widely suffered. Both sarcopenia and dementia are a severe burden for older people and both diseases are hallmarks of frailty. Sarcopenia and its consequences (weakness, slowness, reduction of physical activity, and weight loss) are essential features of “physical frailty” [1]. Whereas cognitive impairment is a characteristic feature of “cognitive frailty”, this is a recently defined condition in which physical frailty coexists with cognitive impairment [2]. As both sarcopenia and dementia significantly contribute to frailty, herein we review the impact of hypovitaminosis D on these two diseases.

With the term “sarcopenia”, we indicate a condition characterized by both reduced muscle mass and muscle strength (“dynapenia”). These declines lead to an impairment in physical performance [3]. Due to the heterogeneity of the criteria used to diagnose sarcopenia, with different working groups providing their own definitions, it is difficult to compare the results of different studies. Table 1 summarizes the criteria used for the diagnosis of sarcopenia.

One of the most used algorithms in clinical practice is the algorithm recently developed by the European Working Group on Sarcopenia in Older People (EWGSOP) [4] (Figure 1).

According to the EWGSOP definition, the estimated prevalence of sarcopenia is 10–40% amongst elderly individuals in the community [5,6], however, this prevalence rises in settings of care for older patients. Due to increasing life expectancy, the number of patients living with sarcopenia is expected to grow to more than 200 million in the next 40 years [3].

Being sarcopenic increases sanitary costs and decreases quality of life in older patients [12,13], hence the correct identification of risk factors linked to sarcopenia is important to precociously identify these patients and to set up an early intervention. In our “aging world”, cognitive impairment is also constantly increasing. The World Alzheimer Report 2018 [14] states that about 50 million people worldwide suffer from dementia and the epidemiological projection envisions that these numbers are expected to rise to 152 million by 2050.

The most diffuse form of dementia is Alzheimer’s disease (AD). Other types of dementia are vascular dementia, mixed dementia, and Lewy body dementia. Dementia and sarcopenia are responsible for an enormous increase in sanitary costs; the United States spent about $818 billion for patients with dementia in 2015, and these costs have increased by more than 30% since 2010. The cost distribution is not homogeneous and the majority of costs burden high-income countries [15].

Both sarcopenia and dementia have been associated with hypovitaminosis D, which has been suggested as one of the causative mechanisms. Although different values have been used as references and the agreed threshold for the diagnosis of hypovitaminosis D is still debated, hypovitaminosis D is widely diffuse among older people. Guidelines from different scientific societies and different countries establish the threshold for hypovitaminosis D as being under 50 nmol/L or 75 nmol/L of blood 25(OH) vitamin D (25(OH)D3) [Table 2].

Despite these different thresholds, the majority of studies indicate that with 25(OH)D3 lower than 50 nmol/L, bone metabolism is impaired and there is an increased risk of fractures, falls, and myopathy [20,21]. Priemel and colleagues demonstrated that pathological mineralization defects of bone occur in patients with a serum 25(OH)D3 below 75 nmol/L [22]. In contrast with the above mentioned studies, two recent randomized controlled clinical trials [23,24] indicated that only individuals with baseline 25(OH)D3 levels lower than 30 nmol/L will benefit from vitamin D supplement use.

Clinicians working on vitamin D generally agree to maintain 25(OH)D3 levels between 20 and 125 nmol/L. These prudential levels are associated with a healthy skeleton and avoid possible toxic effects. Recent papers have raised caveats concerning toxic effects of high doses of vitamin D: the administration of a bolus of vitamin D3 higher than 50,000 UI may result in an increased risk of falls and fractures [25,26]. Moreover, levels of vitamin D higher than 150 nmol/L have been associated with increased mortality in a wide population study [27].

Hypovitaminosis D has been described as common to different chronic diseases linked to senescence, and the incidence of hypovitaminosis D increases as age increases.

Even adopting the most conservative cut-off value for hypovitaminosis D (lower than 50 nmol/L), it is a frequent condition amongst people aged 65 years or more. In the U.S., the prevalence of vitamin D deficiency and vitamin D insufficiency were found to be 28.9% and 41.4%, respectively [28].

Hypovitaminosis D in older age groups is mainly due to the reduced ability of the skin to synthetize cholecalciferol from its precursor: 7-dehydrocholesterol. Together with a reduced synthesis of vitamin D, older subjects showed a reduced expression of vitamin D receptors (VDRs). These two phenomena cooperate in the amplification of the effect of hypovitaminosis D during aging [29].

Hypovitaminosis D has been identified as a common feature between diseases widely diffused in senescence such as osteoporosis, sarcopenia, and cognitive impairment.

Although the most well-known clinical effect of hypovitaminosis D is osteoporo-malacia, here we will focus on the role of vitamin D in the pathogenesis of sarcopenia and cognitive impairment, analyzing studies from experimental models and their clinical relevance.

## 2. Vitamin D and Aging

### 2.1. Vitamin D and Muscle Health: From Bench to Bedside

VDRs are expressed in human muscle fibers, especially during their early developmental phases, and decrease upon their maturation [30]. It has been demonstrated in vitro that vitamin D plays an active role in muscle maturation as myoblasts can differentiate into myocytes thanks to a signal mediated by VDRs [31]. Besides its genomic effects, vitamin D has non-genomic rapid-onset effects that can play a role in muscle contraction as it is involved in the regulation of membrane calcium channels [32,33]. Vitamin D increases calcium influx in muscle cell cytoplasm within minutes in a dose-dependent manner [34,35] through activation of two kinases, namely, c-Src and PI3K [36]. PI3K activation leads to an increasing level of inositol triphosphate (IP3) and diacylglycerol (DAG); IP3 induces calcium displacement from the sarcoplasm, while DAG, along with calcium in the cytosol, is a key component in the activation of protein kinase C (PKC). PKC interacts with calcium channels on the cell membrane, leading to more calcium influx in the cytosol [37,38]. Calcium binds to the troponin–tropomyosin complex, resulting in the exposure of active binding sites, enabling muscle contraction [39].

Experimental models of knock-out mice for VDR confirmed this role, as in VDR null mice, the muscle mass is reduced and the fibers have a lower diameter in respect to wild-type littermates [40].

Moreover, muscle development and maturation is impaired in VDR null mice. These mice express transcriptional factors typical of early muscle development, such as myf5, E2A, and myogenin, for a longer period, suggesting that the VDR is a key player in correct muscle growth and maturation [40].

The action of vitamin D in muscle development is independent of its effect on blood calcium levels, as has been demonstrated in VDR null mice with normal blood calcium [40].

Besides the role on muscle development and maturation, a role for vitamin D has also been postulated in the control of muscle atrophy. Vitamin D has been implicated in muscular protein degradation through the control of the ATP-ubiquitin-dependent system [41]. In rats, a significant increase in catalytic activity and in protein ubiquitination during vitamin D deficiency have been demonstrated [41]. The increased muscle atrophy associated with vitamin D deficiency is associated with reduced anaerobic capacity and tolerance to exercise, together with a disruption of muscle morphology demonstrated by low cross-sectional areas of muscle fibers and the reduction of fast fibers [42].

As VDR expression increases after muscular injury, it has also been suggested that vitamin D may have a role in muscle regeneration [32,43]. This could be very important in patients affected by sarcopenia who have reduced muscular regeneration.

Taken together, these experimental data strongly suggest a role for vitamin D in muscle health. As regards humans, despite studies’ heterogeneity, a physiological role for vitamin D has been envisioned.

In humans, the role of vitamin D in muscle health is supported by several data: in patients with VDR mutations or severe vitamin D deficiency, there is a generalized muscle atrophy and muscular sufferance which appears even before the appearance of the altered bone turnover [43]. Associated studies have demonstrated that in older age, vitamin D deficiency is strongly linked to muscle weakness and loss, suggesting that hypovitaminosis D in the elderly may be an important factor in the development of sarcopenia [39,44,45]. In a large study on more than 4000 older community-dwelling adults, Aspell et al. showed that patients with vitamin D lower than <30 nmol/L were more likely to have impaired muscle function with a reduction in physical performance and muscle strength, but not an increased risk of falls [46].

### 2.2. Vitamin D Supplements and Muscle Health

Despite the accumulating evidence on the link between vitamin D deficiency and muscle health (especially in older adults), the role of vitamin D supplementation in recovering muscle mass and function is yet to be proven. Meta-analysis and systematic literature reviews were able to find only a minor, often non-statistically significant, improvement in muscle strength with vitamin D supplementation, even when associated with calcium supplementation and exercise [47,48].

A closer look at recent randomized controlled clinical trials, even ones included in the aforementioned papers, demonstrates a great deal of heterogeneity in patients, levels of vitamin D at baseline, doses of vitamin D supplementation, and even in tests used to measure muscle strength and sarcopenia. While it is true that the older population is itself a very heterogeneous group, the selection of a more precise section of the population might help in finding results that are more suitable to clarify whether vitamin D supplementation has a role in the prevention and treatment of age-related muscle loss.

Moreover, some trials have raised some caveats suggesting that a high bolus of cholecalciferol does not prevents falls, but, on the contrary, seems to increase the risk of falling and may be ineffective in improving bone mineral density and bone turnover [26].

In order to understand these conflicting results, it is important to highlight that the subjects included in those papers were not affected by hypovitaminosis D and were treated with doses much higher than the ones recommended in clinic [19]. Hence, the conclusions from those papers can only be that too much vitamin D, if it is not needed, may be detrimental for unknown reasons.

Although calcium is important for muscle contraction, randomized clinical trials show no additional effect of calcium on muscle strength in young athletes both female [49] and male [50]. Furthermore, in older community-dwelling women, the administration of yogurts fortified with vitamin D (200 IU) and calcium (400 mg) given twice a day was not able to increase gait performance [51].

### 2.3. Vitamin D and Cognitive Health: From Bench to Bedside

The observation that the VDR is also expressed in the central nervous system (CNS) and that CNS is per se able to synthetize calcitriol thanks to the expression of the 25-hydroxylase and 1α-hydroxylase [52] has raised the hypothesis that vitamin D may have a role in brain health and in cognitive performance.

In a rat model of Alzheimer’s disease (AD), rats fed with low levels of vitamin D lost their cognition abilities more rapidly in respect to those fed a control diet [53]. Moreover, in mice with low levels of vitamin D, the production of amyloid-β (AB) is increased and there is an increased formation of amyloid plaques, as is typically observed in patients affected by AD [54]. In transgenic mice who spontaneously accumulate AB and develop AD, a diet supplemented with cholecalciferol is able to reduce the amyloid plaque formation by enhancing the amyloid clearance. Consistent with histological changes, vitamin D supplements ameliorate animal cognitive performance [55,56].

The mechanisms through which vitamin D reduces the accumulation of AB and the formation of amyloid plaque are not completely clear. It has been suggested that vitamin D increases the AB clearance by the blood–brain barrier, increasing its brain-to-blood efflux though both genomic and non-genomic action [57,58]. Moreover, in vitro primary cultures of cortical neurons show that vitamin D is directly implicated in the production of Aβ and can downregulate its expression [59]. Several genes involved in the pathogenesis of AD have a vitamin D responding element within their sequences [60]. These genes are deregulated if vitamin D deficiency occurs during growth [61,62]. However, it has never been demonstrated that hypovitaminosis D during growth may influence cognitive performance in adult life.

Despite all the data obtained from in vitro and from animal models, the role of vitamin D in cognition is far from being elucidated. Its role is probably complex and mediated through the cross talk with other factors such as estrogen and insulin [56]. Transcriptomic analysis of the neocortex of healthy mice and those affected by AD showed that after vitamin D treatment there is a deregulation of pathways related to inflammation and immune response, neurotransmission, vascular processes, and hormonal alterations, suggesting a complex and multiple role for vitamin D rather than a single one in the development of dementia [56].

In humans, some studies have suggested that lower levels of vitamin D are associated with poorer cognition in patients affected by cognitive problems, however a faster loss of cognitive performance has not been demonstrated [63,64]. Furthermore, in older subjects complaining of memory deficits without a diagnosis of dementia, hypovitaminosis D has been associated with lower cognitive performance, namely, poorer mental flexibility [65]. On the other hand, a recent study performed with the Mendelian randomization method found no evidence that hypovitaminosis D may be a cause of cognitive deficits in mid- or later life [66].

Due to the high heterogeneity of the methods used and of the population analyzed, it is difficult to find homogeneous results. Nevertheless, a recent meta-analysis including both patients with impaired cognition and healthy subjects suggests that poorer vitamin D status is associated with poorer cognitive performance with respect to high vitamin D levels, however, the author acknowledges possible publication biases [67].

Data obtained in animals and associated studies in humans have brought substantial attention to the possible role of vitamin D supplements in preventing cognitive decline in humans. However, the data obtained in different studies are difficult to interpret and do not clearly highlight the role of vitamin D in the pathogenesis and treatment of dementia. Interventional studies are less common than observational ones, and results are more inconclusive because the protocols used for supplementation and the population included are largely different. The durations of treatment vary from a single dose to 18 weeks, with different dosages and different molecules. Dhesi and colleagues [68] used ergocalciferol as a single intramuscular injection of 600,000 IU, Przybelski and colleagues used oral ergocalciferol of 50,000 IU three times a week for four weeks [69], Dean and colleagues used oral cholecalciferol capsules of 5000 IU daily [70], whereas Pettersen used oral cholecalciferol in two different doses of 4000 IU versus 400 IU daily for 18 weeks [71]. Moreover, the population enrolled was different with regards to age: old subjects [72] versus young [47]. The settings are also different: community-dwelling residents versus nursing home residents [73,74]. The vitamin D status at inclusion in the study is also different: in the study by Dhesi and colleagues [68], the subjects included were vitamin D deficient, while in the others they could be deficient or not [69,70]. In addition, the study designs are different, as two studies used a placebo-controlled group [68,70] and others did not [69,71].

Recent meta-analyses reviewed three of the four interventional studies cited [68,70] and found that 314 subjects saw no significant benefit in patients treated with vitamin D supplementation [67]. On the other hand, the study by Pettersen suggested an effect of high doses of cholecalciferol (4000 IU/d) in the amelioration of visual memory in heathy adults, particularly among those who had low levels of vitamin D at their enrolment in the trial.

As regards the supplementation of vitamin D together with calcium, a recent randomized controlled clinical trial in healthy older females showed that yogurts fortified with low doses of vitamin D (200 IU) and calcium (400 mg) given twice a day helped in the maintenance of cognitive performance [51].

Taken together, these data suggest that although vitamin D may have a role in the development of the brain and in cognition, data obtained by intervention studies are not sufficient to indicate that the administration of vitamin D, even at high doses, may be useful for patients with impaired cognition. Data on the use of calcium supplements associated with vitamin D are not sufficient to recommend the double supplementation, and further studies are needed to clarify these indications.

## 3. Conclusions

Progressive increases in life expectancy are associated with a constant increase of chronic diseases associated with age that have a great impact on patients’ quality of life and consistently increase sanitary costs as well as the risks of sarcopenia and dementia.

Hypovitaminosis D is associated with aging and with these chronic diseases. Even though a clear pathogenic mechanism is still not elucidated, association studies have shown a relationship between low vitamin D levels and sarcopenia and dementia. However, interventional studies are not sufficient to recommend treatment with vitamin D to efficiently treat these two diseases.

## Figures and Tables

**Figure 1 nutrients-12-00628-f001:**
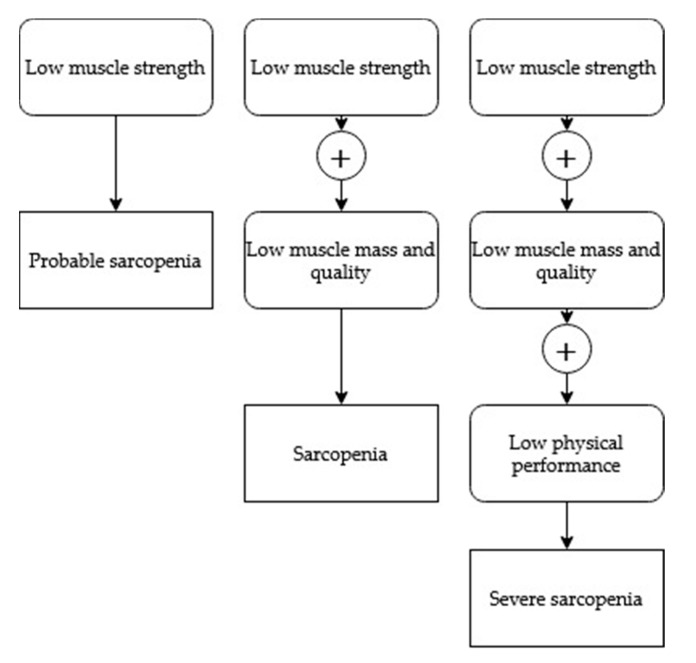
EWGSOP clinical practice algorithm for the diagnosis of sarcopenia.

**Table 1 nutrients-12-00628-t001:** Different criteria for the diagnosis of sarcopenia.

Criterium	Slowness	Weakness	Low Lean Mass	Summary Definition
International Working Group [7]	Gait speed < 1.0 m/s	Not included	ALM/ht^2^ ≤ 7.23 kg/m^2^	Sarcopenia: slowness and low lean mass
EWGSOP [8]	Gait speed ≤ 0.8 m/s	Grip strength < 30 kg	ALM/ht^2^ ≤ 7.23 kg/m^2^	(1) Sarcopenia: low lean mass plus slowness or weakness
				(2) Severe sarcopenia: all three criteria
FNIH Sarcopenia Project primary definition [9]	Gait speed ≤ 0.8 m/s	Grip strength < 26 kg	ALM/body mass index < 0.789	(1) Weakness and low lean mass
				(2) Slowness with weakness and low lean mass
Baumgartner [10]	Not included	Not included	ALM/ht2 ≤ 7.23 kg/m^2^	Low lean mass
Newman [11]	Not included	Not included	Residual of actual ALM*-predicted ALM from equation	Low lean mass

* ALM: appendicular lean mass.

**Table 2 nutrients-12-00628-t002:** Different vitamin D cut-offs used in clinical practice.

	Vitamin D		
Source	Deficiency	Insufficiency	Sufficiency
IOM (2010) [16]	30 nmol/L	30–50 nmol/L	75–250 nmol/L
U.S Endocrine Society (2011) [17]	50 nmol/L	50–75 nmol/L	75–250 nmol/L
SIOMMS (2016) [18]	<25 nmol/L	25–50 nmol/L	50–80 nmol/L
AME (2018) [19]			>75 nmol/L in patients at risk of bone disease

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
