# Peer review of "Hypovitaminosis D and Aging: Is There a Role in Muscle and Brain Health?"

_nutrients, 2020, doi:10.3390/nu12030628_

Round 1

Reviewer 1 Report

line 9: older population not populations please check

lines 16 and 17 are repetition of line lines 9 and 10 please correct

line 162 please check trough ? may be you want to write through.

Author Response

Thank you for your input. Corrections have been made to the paper. Regards

Reviewer 2 Report

Thanks for the comprehensive review about the relevance of vitamin D in muscle and brain health in aging. The overview of the interventions describes supplementation with vitamin D alone in different target groups. Since nutritional benefits are rarely based on only one nutrient, I would like to see some perspective of vitamin D supplementation in the interplay with other nutrients (in diet or supplementation) to preserve muscle and brain health.  

Author Response

thank you for your feedback, corrections have been done. regards
